# Safety and Efficacy of Erythrocyte Encapsulated Thymidine Phosphorylase in Mitochondrial Neurogastrointestinal Encephalomyopathy

**DOI:** 10.3390/jcm8040457

**Published:** 2019-04-05

**Authors:** Michelle Levene, Murray D. Bain, Nicholas F. Moran, Niranjanan Nirmalananthan, Joanna Poulton, Mauro Scarpelli, Massimiliano Filosto, Hanna Mandel, Andrew D. MacKinnon, Lynette Fairbanks, Dario Pacitti, Bridget E Bax

**Affiliations:** 1Molecular and Clinical Sciences, St. George’s, University of London, London, SW17 ORE, UK; mlevene@sgul.ac.uk (M.L.); mdbain@sgul.ac.uk (M.D.B.); 2Department of Neuroscience, East Kent Hospitals Foundation Trust, Canterbury, CT1 3NG, UK; nickmoran@nhs.net; 3Departments of Neurology and Neuroradiology, Atkinson Morley Regional Neurosciences Centre, St George’s Hospital, London, SW17 0QT, UK; n.nirmalananthan@nhs.net (N.N.); andrew.mackinnon@stgeorges.nhs.uk (A.D.M.); 4Nuffield Department of Obstetrics and Gynaecology, University of Oxford, Oxford, OX3 9DU, UK; joanna.poulton@obs-gyn.ox.ac.uk; 5Neurology Unit, Azienda Ospedaliera Universitaria Integrata Verona, Piazzale Aristide Stefani, 1, 37126 Verona, Italy; mauro.scarpelli@aovr.veneto.it; 6Center for Neuromuscular Diseases, Unit of Neurology, ASST Spedali Civili and University of Brescia, 25100 Brescia, Italy; massimiliano.filosto@unibs.it; 7Galilee Medical Center, Nahariya 22100, Israel; hanam2@gmc.gov.il; 8The Purine Research Laboratory, St Thomas’ Hospital, London SE1 7EH, UK; Lynette.Fairbanks@viapath.co.uk; 9College of Medicine and Health, St Luke’s Campus, University of Exeter, EX1 2LU, UK; d.pacitti@exeter.ac.uk

**Keywords:** Mitochondrial neurogastrointestinal encephalomyopathy, Mitochondrial neurogastrointestinal encephalomyopathy (MNGIE), nuclear thymidine phosphorylase gene (*TYMP*), enzyme replacement, thymidine phosphorylase, mitochondrial disease, rare disease, orphan disease

## Abstract

Mitochondrial neurogastrointestinal encephalomyopathy (MNGIE) is an ultra-rare autosomal recessive disorder of nucleoside metabolism that is caused by mutations in the nuclear thymidine phosphorylase gene (*TYMP*) gene, encoding for the enzyme thymidine phosphorylase. There are currently no approved treatments for MNGIE. The aim of this study was to investigate the safety, tolerability, and efficacy of an enzyme replacement therapy for the treatment of MNGIE. In this single centre study, three adult patients with MNGIE received intravenous escalating doses of erythrocyte encapsulated thymidine phosphorylase (EE-TP; dose range: 4 to 108 U/kg/4 weeks). EE-TP was well tolerated and reductions in the disease-associated plasma metabolites, thymidine, and deoxyuridine were observed in all three patients. Clinical improvements, including weight gain and improved disease scores, were observed in two patients, suggesting that EE-TP is able to reverse some aspects of the disease pathology. Transient, non-serious adverse events were observed in two of the three patients; these did not lead to therapy discontinuation and they were managed with pre-medication prior to infusion of EE-TP. To conclude, enzyme replacement therapy with EE-TP demonstrated biochemical and clinical therapeutic efficacy with an acceptable clinical safety profile.

## 1. Introduction

Mitochondrial neurogastrointestinal encephalomyopathy (MNGIE) is an ultra-rare autosomal recessive metabolic disorder (Online Mendelian inheritance in Man #603041, Genome Database accession #9835128) that is almost universally fatal [1]. It causes relentless and progressive morbidity, followed by premature death at an average age of 37.6 years [2]. MNGIE is caused by mutations in the nuclear thymidine phosphorylase gene (*TYMP*) and a subsequent deficiency in thymidine phosphorylase activity (EC 2.4.2.4) [3,4]. The thymidine phosphorylase enzyme plays a pivotal role in the deoxyribonucleoside salvage metabolic pathway through catalysing the reversible phosphorylation of thymidine (also known as deoxythymidine) and deoxyuridine to 2-deoxyribose 1-phosphate and thymine and uracil, their respective bases [3]. There is a systemic accumulation of thymidine and deoxyuridine in the absence of thymidine phosphorylase activity, which then generates imbalances within the mitochondrial deoxyribonucleotide pools, causing mitochondrial DNA (mtDNA) point mutations, depletion and deletion abnormalities, and ultimately mitochondrial dysfunction [5,6,7,8,9]. Gastrointestinal dysfunction with frequent intestinal sub-occlusive episodes, malnutrition, cachexia and neurological involvement, including ptosis and ophthalmoparesis, peripheral neuropathy, myopathy, symmetric distal weakness, which prominently affects the inferior extremities, and leukoencephalopathy mainly typify the clinical picture of MNGIE [3,5,10].

No specific therapies for patients with MNGIE have been established in clinical trials. However, a number of experimental therapeutic approaches have been investigated, which either directly remove the deoxyribonucleosides (haemodialysis and peritoneal dialysis) or introduce the deficient thymidine phosphorylase (platelet transfusions, allogeneic haematopoietic stem cell transplantation (AHSCT), autologous erythrocyte-encapsulated thymidine phosphorylase (EE-TP), and orthotopic liver transplantation) [6,11,12,13,14,15,16,17,18,19,20].

EE-TP is under clinical development by us and it is formulated by the ex vivo encapsulation of *Escherichia coli* (*E.coli*) thymidine phosphorylase within the patient’s autologous erythrocytes while using a reversible hypo-osmotic dialysis process. The enzyme-loaded erythrocytes are then returned to the patient. The rationale for this therapeutic approach is based on the facilitated diffusion of thymidine and deoxyuridine across the erythrocyte membrane via ENT1, the equilibrative nucleoside transporter. Once inside the erythrocyte, the encapsulated enzyme catalyses the metabolism of the deoxyribonucleosides to the normal products, thymine and uracil, which then exit the erythrocyte and enter the normal metabolic pathways. EE-TP aims to normalise the systemic accumulations of thymidine and deoxyuridine, and thereby ameliorate the mitochondrial deoxyribonucleotide pool imbalances and slow disease progression. EE-TP has the pharmacological advantages of prolonging the circulatory half-life of the enzyme and potentially minimising the immunogenic reactions that are frequently observed in enzyme replacement therapies that are administered by the conventional route [18,19,21].

Here, we describe three adult patients with MNGIE (Patients 1 to 3) who were treated with EE-TP under a compassionate use programme. Clinical and metabolic improvements were previously reported in Patient 2, who was treated with EE-TP for 27 months [19]; here, we provide an update on the subsequent 49 months of treatment in this patient. EE-TP was shown to decrease the plasma metabolites in all three patients and clinical improvements, including weight gain and improved disease scores, were observed in two patients. 

## 2. Experimental Section

*Study participants*. Female and male patients ≥ 18 years and of all ethnic backgrounds, on a world-wide basis were considered for study inclusion based on the following defined inclusion criteria: (i) a definitive diagnosis of MNGIE due to thymidine phosphorylase deficiency that as established by the confirmation of a pathogenic mutation in *TYMP*, <18% of normal thymidine phosphorylase activity in the buffy coat, and plasma thymidine and deoxyuridine concentrations greater than 3 µmol/L and 5 µmol/L, respectively; (ii) a whole blood haematocrit ≥20%; and, (iii) patient signed informed consent. The exclusion criteria were: (i) pregnancy, planned pregnancy, breast feeding, or unwillingness to practice contraception, (ii) participation in a controlled trial of an investigational medicinal product within the previous 12 weeks, (iii) patients who had received a successful bone marrow transplant; and, (iv) patients with a known history of human immunodeficiency virus, hepatitis B infection, or an active hepatitis C infection.

*Study design and objectives*. This study was conducted in an open label manner with the participants receiving EE-TP in accordance with the provisions of Schedule 1 of The Medicines for Human Use (Marketing Authorisations etc.) Regulations SI 1994/3144, where Schedule 1 provides an exemption from the need for a marketing authorisation for a relevant medicinal product, which is supplied on an individual patient basis to fulfill a "special need". The study objectives were to assess the safety, tolerability, and efficacy of EE-TP when they were administered to patients with MNGIE at dose-escalating intravenous infusions. 

After a 28 day screening and baseline assessment period, the participants received their first intravenous infusion of EE-TP. For safety reasons, the study was initiated with the recruitment of a single patient (Patient 1), who was administered a low dose of EE-TP (dose level 1, 4 U/kg). This dose was labelled with sodium [^51^Cr] chromate for the purpose of evaluating the in vivo survival characteristics of the thymidine phosphorylase-loaded erythrocytes. In subsequent treatment cycles, the administered dose of EE-TP was cautiously escalated with the aim of achieving metabolic correction, provided that the patient successfully completed the safety reviews between administrations. For Patient 1, the dose was escalated in treatment cycle 2 to level 2 (9 ± 2 U/kg) and this was followed by dose escalations in treatment cycles 4 (dose level 3, 18 ± 2 U/kg) and 9 (dose level 4, 29 ± 2 U/kg), Figure 1. This dose escalation strategy was applied to Patients 2 and 3, but without sodium [^51^Cr] chromate-labelling, and while employing higher starting doses, based on the safety and tolerability assessments that were made on Patient 1. Patient 2 initiated treatment with dose level 3 and this was escalated to dose level 4 in treatment cycle 3 and then to dose level 5 (47 ± 2 U/kg) in treatment cycle 24. Patient 3 initiated treatment with dose level 5, and this was escalated to dose level 6 (108 ± 5 U/kg) in treatment cycle 3. See Figure 1 for the dose escalation scheme that was employed for Patients 1, 2, and 3.

*Preparation and administration of EE-TP*. The study investigational product, EE-TP was prepared by the investigator team, using sterile, single-use materials, and reagents throughout, under a Specials licence held by St. George’s Healthcare Trust Pharmacy according to the Rules and Guidance for Pharmaceutical Manufacturers 2007 (MHRA), within Class A isolators contained within a Pharmacy Clean room. EE-TP was formulated by the encapsulation of thymidine phosphorylase within the patient’s own autologous erythrocytes, ex-vivo, while using a reversible hypo-osmotic dialysis process [21]. Briefly, a predetermined volume of the patient’s blood was collected (55 to 400 mL; the volume collected being dependent on the activity of thymidine phosphorylase to be encapsulated and the patient’s haematocrit) using aseptic techniques into tubes containing dalteparin sodium, the heparin with low molecular weight (10 units ⁄mL blood). Within cleanroom facilities, the blood components were separated by centrifugation at 1100× *g* for 10 min, followed by the removal of the plasma and buffy coat (both retained for later use). The erythrocytes were then washed twice in cold (4 °C) phosphate buffered saline (PBS; 136.89 mmol/L NaCl, 2.68 mmol/L KCl, 8.10 mmol/L Na_2_HPO_4_, 1.47 mmol/L KH_2_PO_4_, pH 7.4) with centrifugation. Seven volumes of washed and packed erythrocytes were mixed with three volumes of cold PBS that contained 6000 to 38,000 Units recombinant *E. coli* thymidine phosphorylase (Diatheva, Italy for treatment cycles 68–79 in Patient 2, otherwise Sigma, Israel) and the suspension placed into dialysis bags with a molecular weight cut-off of 12,000 Daltons. The cells were dialysed against hypo-osmotic buffer (5 mmol/L KH_2_PO_4_, 5 mmol/L K_2_HPO_4_, pH 7.4) at 4 °C in a refrigerated incubator with rotation at 6 rpm for 120 min. These conditions of hypo-osmolarity induce the swelling of erythrocytes due to an influx of water, until at a critical size, pores form in the erythrocyte membrane; whilst, permeable thymidine phosphorylase enters the erythrocytes by diffusion. Erythrocyte resealing was achieved by the restoration of iso-osmotic conditions by transferring the dialysis bags to pre-warmed iso-osmotic PBS that was supplemented with 5 mmol/L adenosine, 5 mmol/L glucose, and 5 mmol/L MgCl_2_, pH 7.4, and rotated at 6 rpm for 60 min in an incubator set at 37 °C. The resealed erythrocytes were washed three times in three volumes of supplemented PBS with centrifugation at 100× *g* for 20 min. The washed and packed EE-TP was mixed with the retained buffy coat and re-suspended in an equal volume of plasma. EE-TP was administered to the patient by slow intravenous infusion within 45 min of manufacture. Figure 2 details the manufacturing steps for EE-TP preparation. 

*In vivo survival studies of EE-TP*. Thymidine phosphorylase-loaded erythrocytes were labelled using a standard ^51^Cr erythrocyte-labelling technique [22]. The washed and packed EE-TP that was prepared as described above was gently mixed with 0.75 MBq of sodium [^51^Cr] chromate (Amersham International, Buckinghamshire, UK), and then allowed to stand at room temperature for 30 min. Unbound chromium was removed by the addition of 100 mg of ascorbic acid (100 mg/mL, Evans Medical, Leatherhead, UK) to the cell suspension, followed by a single wash in supplemented PBS. After resuspension in an equal volume of autologous plasma, EE-TP was slowly injected into patient 1 over a period of 10 min.

In vivo survival was assessed by monitoring the disappearance of ^51^Cr label from the circulation; 10 mL blood samples were taken from a vein in the contralateral arm 3, 6, 8, and 14 days, and then weekly until activity was not noticeably above background. To check for intravascular haemolysis, ^51^Cr activity was measured in plasma. Percentage raw cell survival was calculated by expressing the CPM/mL packed cells as a percentage of the calculated zero time value, after correction for natural decay. The data were then plotted against time on a semi-logarithmic scale and a best-fit line fitted. T_1/2_ thymidine phosphorylase-loaded erythrocyte survival was taken as the time for the concentration of ^51^Cr in the circulating blood to fall to 50% of its initial value. For the determination of mean cell life, CPM/mL packed cells (corrected for both natural decay and chromium elution from the erythrocytes) were expressed as a percentage of the corrected zero time value. The cell survival data were plotted against time on a linear scale and the mean cell life span derived from the intercept that was obtained by extrapolation of the line to zero activity.

Making three 24-h urine collections for the first 72 h after injection assessed the urinary excretion of the label; ^51^Cr activity in the urine was expressed as a percentage of the total ^51^Cr activity injected.

*Patient assessments*: Safety assessments included the documentation of all adverse events, vital signs, and standard clinical laboratory evaluations, including haematology and blood chemistry. Vital signs were monitored for 2 h following the administration of EE-TP, for the first three treatment cycles and after each dose escalation. Clinical laboratory evaluations were performed following the end of each treatment cycle, prior to the administration of EE-TP in the following treatment cycle.

To assess the metabolic efficacy of EE-TP in reducing or eliminating plasma thymidine and deoxyuridine, the disease associated metabolites, 5 mL of whole blood was collected into EDTA vacutainers at pre-therapy baseline (day 0) and then five days following each administration of EE-TP. The assessment of serum anti-thymidine phosphorylase antibodies for immunogenicity assessment was conducted by collecting 4 mL of whole blood into serum separation tubes (SST) vacutainers at pre-therapy baseline (day 0) and then prior to each administration of EE-TP.

The clinical efficacy of EE TP was assessed by following longitudinal changes from baseline in body weight measurements, MRI scan (patient 2 only), disease scoring scales (MRC sum score, Overall neuropathy limitations scale (ONLS), Newcastle mitochondrial disease scale (NMDS), and Sensory sum score), and the Short Form (36) Health Survey (SF36).

*Analytical methods*. Plasma and urine thymidine and deoxyuridine levels were measured while using Ultra-Performance Liquid Chromatography (UPLC). Briefly 200 µL of plasma were mixed with 200 µL 10% TCA, vortexed, and then centrifuged for 2 min at 12,000 rpm. The supernatant was then washed four times with water saturated diethyl-ether, after which the samples were left to stand for 5 min to evaporate any residual ether. The urine samples were prepared by diluting 1:31 with deionised water. Two microliters of processed plasma or urine sample were injected onto a UPLC column. Chromatographic separation of thymidine and deoxyuridine was achieved while using reversed phase chromatography with gradient elution using a Waters Alliance Acquity UPLC system with UV detection. A Waters Acquity BEH C18 1.7 μm column (2.1 × 150 mm) with a Vanguard C18 pre-column was used as the stationary stage. Analytes were eluted using a gradient of Buffer A (40 mM ammonium acetate, pH 5.0) and Buffer B (100% *v*/*v* Methanol), which was set at 100% Buffer A for 3.5 min, followed by a linear gradient to 20% Buffer B at 12 min at a flow rate of 0.2 mL/min. The total elution time was 15 min at 20 °C with UV detection at 254 nm and 0.1 absorbance units full scale. Comparing spectra with pure standards identified metabolites. The retention times of deoxyuridine and thymidine were 9.29 and 11.76 min, respectively.

Thymidine phosphorylase activity was measured in an aliquot of each batch of EE-TP that was manufactured to ascertain the dose to be administered using a validated high performance liquid chromatography (HPLC) assay [23].

Anti-thymidine phosphorylase antibodies were measured using a validated thymidine phosphorylase assay [24]. The samples were screened for a positive or negative signal; for samples that screened positive, the specificity of thymidine phosphorylase was determined in a confirmatory assay with a specificity cut-point of 93% inhibition. 

*Study approval*. The National Research Ethics Service Committee approved the study. The participants provided written informed consent before participating in any study procedures. 

## 3. Results

### 3.1. Patient Characteristics 

Three patients between the ages 25 and 28 years were recruited into the study. The patients fulfilled the recruitment criteria through harbouring pathogenic mutations in the thymidine phosphorylase gene, having a deficiency of thymidine phosphorylase activity in the buffy coat, and having plasma thymidine and deoxyuridine concentrations that were greater than 3 µmol/L and 5 µmol/L, respectively. Table 1 shows the demographic and baseline disease data. Clinical manifestations on recruitment for Patients 1 and 3 were peripheral polyneuropathy, external ophthalmoplegia, intestinal dysmotility, anorexia, and cachexia, and for Patient 2, peripheral polyneuropathy and external ophthalmoplegia with minimal intestinal involvement.

### 3.2. In Vivo Survival of EE-TP

To ascertain that the encapsulation of thymidine phosphorylase had no effect on the in vivo survival of erythrocytes, ^51^Cr labelling estimated the circulatory lifespan of the first EE-TP dose that was administered to patient 1. The mean cell life and mean cell half-life (t_1/2_) were calculated to be 108 and 32 days, respectively, both within the normal reference range (Figure 3) and the mean daily urinary excretion of label was 0.9% (1.1% at 0–24 c, 0.8% at 24–48 h, and 0.7% at 48–72 h), also within the normal ^51^Cr elution limits of 1.0–3.2% per day (Table 2). No label was detected in the plasma demonstrating minimal intra-vascular haemolysis of the enzyme loaded cells.

### 3.3. Efficacy Outcomes

#### 3.3.1. Patient 1

Thirty-one treatment cycles of EE-TP (including the chromium labelling study) were administered over a period of 28 months. EE-TP was administered at ascending dose levels of 4, 9, 18, and 29 U/kg body weight/four weeks up to treatment cycle 24; this was followed the administration of 14 U/kg body weight/two weeks between treatment cycles 25 to 31 (Figure 4a). The pre-treatment plasma concentrations of thymidine and deoxyuridine were 10 µmol/L and 20.0 µmol/L, respectively; between 270 and 620 days from the start of therapy, these were reduced to intra cycle concentrations of 2–6 µmol/L for thymidine and 3–13 µmol/L for deoxyuridine (Figure 4b). From day 320 onwards, the urinary excretion of metabolites decreased from pre-therapy levels of 73 and 118 µmol/24 h for thymidine and deoxyuridine, respectively, to 0–41 µmol/24 h for thymidine and 0–49 µmol/24 h for deoxyuridine (Figure 4c). By 200 days of therapy, the patient had gained 4 kg in weight (Figure 4d). This coincided with a decrease in nausea and vomiting, an increased walking distance, and an increase in the physical and mental components of the SF36 health and well-being survey from 47 and 43, respectively, at pre-therapy, to 52 and 45 at day 189 (seven months). No changes were noted in the disease scoring scales at seven months when compared to the pre-therapy scores (Table 3).

At day 220, the patient developed a flu-like illness and this was followed by a diagnosis of small intestinal bacterial overgrowth, which led to a subsequent 4 kg weight loss (Figure 4d). At day 252 transparenteral nutrition (TPN) was commenced with the administration of 2000 mL of lipid-based formula three times per week (500 Kcal medium-chain triglycerides and 1200 Kcal glucose) and 2000 mL of lipid-free formulation (1400 Kcal glucose) four times per week. Although this resulted in an initial 6 kg increase in weight up to day 686 (Figure 4d), the activities of alanine minotransferase (ALT) and gamma-glutamyl transpeptidase (GGT) became elevated, in a pattern that was indicative of hepatocellular damage (Figure 5). From day 620, the plasma metabolite levels increased to levels that were equal or greater than those determined pre-therapy (Figure 4b) and this correlated with the development of lipaemia and lipiduria in association with the continuation of TPN. There was a progression of the peripheral polyneuropathy and a deterioration in the NMDS general neurological functioning and clinical assessment components, the MRC score, the sensory sum score, and the SF36 health and well-being survey when compared to assessments that were recorded pre-treatment and seven months of treatment with EE-TP (Table 3). In an attempt to lower the metabolite levels, the frequency of treatment cycles was increased to fortnightly (14 U/kg/two weeks) from treatment Cycle 26 onwards. This had no effect on the metabolite levels, and the patient died from general debilitation 20 days after the last administration of EE-TP.

#### 3.3.2. Patient 2

A total of 79 treatment cycles of EE-TP were administered over of period of 76 months. Data that was obtained for the first 27 months of treatment (cycles 1 to 30) is reported elsewhere [19]. Between months 27 and 76, EE-TP doses of 47 U/kg per four weeks (dose level 5) were administered. The pre-treatment plasma concentrations of thymidine and deoxyuridine were 20.5 µmol/L and 30.6 µmol/L, respectively, and these were reduced to intra cycle values of 0–9 µmol/L for thymidine and 0–15 µmol/L for deoxyuridine (Figure 6a). These were lower than the levels that were previously reported for the first 27 months of therapy [19]. The intra-cycle urinary excretion rates of thymidine and deoxyuridine ranged between 0–354 µmol/24 h for thymidine and 4–218 µmol/24 h for deoxyuridine (Figure 6b); these were generally lower than those reported for the first 27 months of therapy [19]. The pre-therapy excretion rates were 421 µmol/24 h for thymidine and 324 µmol/24 h for deoxyuridine, respectively. The previously reported 5.8 kg gain in body weight was sustained for a total of 23 months until day 1162, after which there was a 6 kg weight loss following a flu-like illness. Body weight then increased to 59 kg but it then declined to 55.5 kg after 11 months, following a further flu-like illness (Figure 6c). The formerly reported a decline in plasma creatine kinase activity, from a pre-therapy activity of 1200 U/L to activities that are very near or within the normal reference range of 40–320 U/L, were sustained throughout this current reporting period (Figure 6d). 

The scores for the physical and mental components of the SF36 health and well-being survey increased from 52 and 59, respectively, at 23 months, to 55 and 60 at 50 months of therapy. The scores remained at these levels at 64 and 73 months (Table 4). Clinical assessments 64 months after initiating treatment with EE-TP demonstrated further improvements in sensory ataxia, balance and gait, and fine finger functioning when compared with assessments that were recorded at 23 months of therapy. The first improvement in the general neurological functioning component of the NMDS was recorded at 73 months of therapy, along with an improvement in the clinical assessment component at 64 months and a further improvement at 73 months of therapy. A decline in the system specific functioning component of the NMDS was noted at 64 months of therapy, as compared with the previous recorded scores. However, this was followed by a small improvement at 73 months of therapy. The MRC sum score for power remained unchanged from 23 months, with the patient demonstrating normal power in his right little finger and thumb. The score for the ONLS scale declined at 64 months of therapy, as compared to scores that were recorded at pre-therapy and up to 50 months of therapy. There were improvements of 7 and 16 points on the Sensory sum score, respectively, at 64 and 73 of months of therapy, when compared to the score that was recorded at 23 months of therapy (Table 4). Patient reported outcomes correlated with neurological assessments and they included improvements in the distal sensation in the upper and lower limbs and being able to hold a focused conversation whilst standing, whereas previously his lack of balance made visual engagement difficult. 

Brain MRI at month 28 of therapy showed almost symmetric, patchy areas of FLAIR and T2 hyperintensity in the cerebral periventricular, deep, and subcortical white matter (Figure 7a). A follow-up brain MRI at 63 months showed a progression of T2/FLAIR hyperintense cerebral white matter changes (Figure 7b). Relative sparing of the U fibres and the corpus callosum was noted (Figure 7c, left image). Infratentorially, there was minor symmetrical FLAIR hyperintense signal change in the cerebellar peridentate white matter (Figure 7c, right image).

#### 3.3.3. Patient 3

Four treatment cycles were administered over a period of 83 days. The initial EE-TP dose administered was 43 U/kg and this was escalated to 108 U/kg every four weeks in treatment cycles 3 and 5, respectively (Figure 8a). The pre-treatment plasma concentrations of thymidine and deoxyuridine were 12 µmol/L and 19 µmol/L, respectively; these declined to intra-cycle values of less than 4 µmol/L and 2 µmol/L for thymidine and deoxyuridine, respectively, from 60 days onwards (Figure 8b). By day 3 of treatment, the intra-cycle urinary excretion rates of thymidine and deoxyuridine rapidly declined from 308 and 448 µmol/24 h pre-therapy to levels less than 21 µmol/24 h for deoxyuridine and 30 to 89 µmol/24 h for thymidine (Figure 8c). Longitudinal assessments of disease scoring and the SF36 health and well-being survey were not conducted due to the premature termination of therapy due to the patient’s inability to sustain long distance travel to the United Kingdom (UK) once per month to receive treatment. The patient’s weight remained unchanged from the pre-therapy measure of 31.7 kg.

### 3.4. Safety

Adverse reactions were observed in two of the three patients, Patients 1 and 2 in treatment cycles 1–17 and 1–11, respectively. These were short-lived, occurring with 5 min of infusion, and they are described in Table 5 and Table 6 for Patients 1 and 2, respectively. The reactions were managed by the administration of antihistamine and anti-inflammatory therapy, prior to EE-TP administration, with the addition of anti-emetics in Patient 1. It is noteworthy that the introduction of the highly purified GMP enzyme preparation, which was produced by Diatheva in treatment Cycle 68 (Patient 2), permitted a gradual withdrawal of the pre-medication. However, intravenous hydrocortisone (100 mg) was re-introduced in treatment cycles 77–79 as a precaution after the patient experienced chills three hours after the infusion of EE-TP in the previous treatment cycle.

Immunogenicity, i.e., the presence of anti-thymidine phosphorylase antibodies, was positive in the serum samples from two patients in the screening assay (treatment cycle 8 for Patient 1, and treatment cycles 9 onwards for Patient 2). The confirmatory assay revealed that only the positive samples from Patient 2 were specific for thymidine phosphorylase antibodies. There were no correlations between the plasma levels of deoxyribonucleosides and the appearance of anti-thymidine phosphorylase antibodies. 

No clinically significant alterations in the vital signs were reported. No abnormalities of haematological or clinical chemistry parameters were observed (other than those reported above).

## 4. Discussion

MNGIE is a relentlessly progressive, degenerative disease with a very poor prognosis, and it causes a great deal of morbidity for affected individuals. Gastrointestinal dysmotility, which is caused by degeneration of the alimentary peripheral nervous system, occurs in nearly all patients. The resulting symptoms problems include early satiety, dysphagia, nausea, and vomiting after eating, episodic abdominal distention and pain, diarrhoea, and severe weight loss. Peripheral polyneuropathy leads to distal weakness and sensory symptoms and it may be disabling. Ptosis, external ophthalmoplegia, and hearing loss are also common [1]. At present, there is no regulatory approved treatment to prevent or reverse inexorable clinical deterioration. 

A number of experimental therapies have been explored that share the therapeutic strategy of reducing or eliminating the systemic concentrations of thymidine and deoxyuridine with the ultimate aim of ameliorating intracellular deoxyribonucleotide imbalances and preventing further mtDNA damage [6,11,12,13,14,15,16,17,18,19]. AHSCT and more recently orthotopic liver transplantation have provided the permanent restoration of thymidine phosphorylase activity [14,15,16,17,20]. A retrospective analysis of patients who received AHSCT showed that nine of 24 patients (37.5%) were alive at follow-up, with survival being strongly associated with human a leukocyte antigen match of 10/10 and an absence of liver disease and gastro-intestinal pseudo-obstruction [17]. Death was generally caused by transplantation complications or disease progression, and consequently a published consensus conference proposal recommends that only patients with an optimally matched donor and in good clinical condition should be considered for treatment while using AHSCT [14]. Patients with MNGIE frequently experience delays in attaining the correct diagnosis, during which time their clinical condition often deteriorates; therefore, a majority of patients are ineligible for AHSCT. A clinical trial, which is sponsored by Columbia University, USA is currently recruiting patients with MNGIE to evaluate the safety of AHSCT (NCT02363881). Orthotopic liver transplantation for MNGIE has been reported in two patients without complications and with sustained normalisation of thymidine and deoxyuridine (follow-up 90 days to 18 months) [20]. However, a longitudinal evaluation of data that was collected from additionally transplanted patients is necessary to confirm the efficacy of this treatment approach. 

Here, we report the application of an enzyme replacement therapy for MNGIE, which uses a genetically engineered recombinant *E. coli* thymidine phosphorylase sharing a 40% sequence homology with the human sequence [25]. The enzyme is encapsulated into the patient’s autologous erythrocytes ex vivo to produce EE-TP, which is then administered to the patient. In these studies, the encapsulation process was conducted using a centralised manual manufacturing process taking 7 to 8 h from the time of venesection. Consequently, the patients were required to attend the study site on the day of treatment for venesection in the morning for EE-TP manufacture and then infusion later in the day. Two of the patients travelled from abroad every four weeks to receive treatment. For clinical trials, a decentralised manufacturing process for EE-TP will be employed using an automated device, thereby assuring consistency in the EE-TP drug product and permitting treatment to be expanded to the patient’s locality. Patient 2 was transitioned to the automated EE-TP manufacturing process in treatment cycle 80, along with a forth patient that was recruited into this compassionate programme.

Erythrocyte-mediated enzyme replacement therapy is applicable to disorders where the pathological plasma metabolite is able to permeate the erythrocyte membrane. EE-TP aims to clear the associated MNGIE disease metabolites, thymidine and deoxyuridine, from the extracellular and cellular compartments, leading to an amelioration of the intracellular nucleotide imbalances and ultimately an arrest/reverse of the progression of the clinical disease by the reversal of the mitochondrial dysfunction in MNGIE. The encapsulation within erythrocytes has the benefit of prolonging the circulatory half-life of thymidine phosphorylase, thereby reducing the frequency of EE-TP required, and potentially minimising the immunogenic reactions against the *E. coli* enzyme. Indeed, the viability of the erythrocyte as a vehicle for sustaining therapeutic blood levels of thymidine phosphorylase was demonstrated in Patient 1, where the first dose of EE-TP was labelled with ^51^Cr. The circulating mean cell life and mean cell half- life were 108 days and 32 days, respectively, and they were well within the normal mean cell life range of 89 to 131 days and mean cell half-life range of 19 to 29 days [22]. The absence of ^51^Cr in the plasma and the finding that the urinary excretion of ^51^Cr was within the normal Cr elution limit of 1.0 to 3.2% per day indicate that the label was cell associated and there was minimal intravascular haemolysis of EE-TP on infusion.

We investigated the safety, tolerability, and efficacy of EE-TP in three adult patients with MNGIE. The findings from the initial 27 months of treating Patient 2 with EE-TP were previously published and so here we report an update on the subsequent 49 months of treatment of this patient with EE-TP [19]. Six dose levels of EE-TP were investigated, ranging from 4U/kg to 108 U/k every four weeks. A cautious escalation strategy was employed by staggering recruitment, starting with Patient 1, who was administered dose levels 1 to 4 (4–29 U/kg/4 weeks), followed by Patient 2, who was administered dose levels 3 to 5 (18–47 U/kg/4 weeks), and then Patient 3, who was administered dose levels 5 and 6 (47 and 108 U/kg/4 weeks). The decision to escalate to the next dose was based on the patient successfully completing a safety review between EE-TP administrations and the objective of further reduction of the disease plasma metabolites, thymidine, and deoxyuridine. Although dose levels of 4 and lower provided some lowering of the plasma metabolite concentrations, dose levels 5 and 6 provided superior reductions in Patients 2 and 3, with intra-cycles plasma thymidine and deoxyuridine often decreasing to below the diagnosis concentrations for MNGIE. Dose levels of 4 and above were able to significantly reduce the urinary excretion of thymidine and deoxyuridine in all three patients. 

Minor clinical improvements were observed in Patient 1 and more substantial improvements in Patient 2. Both of the patients demonstrated several periods of weight gain, but also periods of weight loss that are associated with intercurrent illnesses. TPN was commenced in Patient 1 and, although this resulted in an initial increase in weight, it was subsequently followed by weight loss, which coincides with a sharp increase in plasma metabolites, an elevation of the liver enzymes ALT, and GGT and the development of hyperlipaemia and hyperlipiduria. A change of EE-TP dosing from 29 U/kg/4 weeks to 14 U/kg/2 weeks failed to reduce the metabolites, which increased to levels that were higher than those recorded pre-therapy. Studies have shown that liver disease can induce pathological alterations in the structure and function of the erythrocyte membrane, and TPN administration can generate changes in the lipid composition and hence the fluidity of the erythrocyte membrane [26,27,28]. Correct membrane function plays a pivotal role in modulating the activity of membrane channels and transporters, and we therefore hypothesize that the lipid component of TPN interfered with the efficacy of EE-TP in Patient 1 at the level of the erythrocyte nucleoside transporter, ENT1. Hepatic steatosis and elevated transaminases, in association with TPN administration, has previously been reported in patients with MNGIE, and also disease-related hepatopathies, including hepatic steatosis, hepatomegaly, increased transaminases, abnormal liver function, triglyceride hyperlipidemia, and cirrhosis [29,30]. Therefore, the administration of TPN should be carefully considered in such patients, as the lipid component is metabolised by the mitochondrion, the site of the primary pathology in MNGIE. Thymidine phosphorylase is highly expressed in the liver of healthy individuals; the administration of TPN should be cautiously appraised for patients with MNGIE who express residual enzyme activity.

The clinical assessments of Patient 2 between 27 and 76 months showed further improvements since the previous reporting period (0 to 27 months). Improvements in sensory ataxia, balance and gait, and fine finger movements were recorded and patient reported outcomes included improvements in the distal sensation in hands and feet. Plasma creatine kinase, a marker of muscle damage, continued to decrease, and remain within the normal range, indicating that EE-TP was able to relieve the skeletal muscle from the toxic effects of the accumulated thymidine and deoxyuridine. These results, and the clinical improvements that were noted in patients who received AHSCT, indicate that systemic reductions in thymidine and deoxyuridine concentrations are able to reverse the somatic mtDNA mutations that accumulate in post-mitotic cells [17]. An early normalization of nucleosides is essential in avoiding the cumulative effects of metabolite imbalances on mtDNA and to prevent as much mitochondrial damage as possible. Therefore, we propose an expeditious treatment with EE-TP. This therapeutic approach would also increase patients’ eligibility for permanent treatments, for example, with AHSCT or OLT, once a match is identified, or for future promising gene therapies [31,32]. Pivotal to this is an increased clinical awareness of MNGIE to enable an early diagnosis before disease progresses to an irreversible state.

Leukoencephalopathy is a hallmark of MNGIE and in the majority of affected individuals, it is initially patchy, but it progressively becomes more diffuse, appearing as hypointense on T1- and hyperintense on T2- weighted images and in FLAIR and fast spin echo (FSE) T2 sequences (1). A follow-up brain MRI scan of Patient 2 showed a progression of T2/FLAIR hyperintense cerebral white matter changes. Therefore, EE-TP therapy was not able to halt the progression of the leukoencephalopathy. However, it is not known whether therapy was able slow down the rate of progression. An additional consideration of the extent and distribution of leukoencephalopathy is not distinctly correlated with age and the clinical phenotype [33]. More extensive MRI studies of untreated and treated patients with MNGIE are required to establish whether the leukoencephalopathy can be reversed. *Post mortem* histopathological studies of brains have shown the presence of albumin in the cytoplasm of reactive astrocytes in patients with MNGIE, as compared to age-matched healthy individuals, suggesting functional blood brain barrier alterations and consequent vasogenic oedema as a cause of leukoencephalopathy in MNGIE [34]. Furthermore, mild perivascular gliosis was also observed in immunohistochemical analyses [35]. Thymidine phosphorylase is also referred to as gliostatin, and it has been shown to exert strong inhibitory effects on glial cells and neurotrophic effects on cortical neurons [36,37,38]. Therefore, the leukoencephalopathy in MNGIE may be a consequence of the absence of the inhibitory activity of thymidine phosphorylase, rather than its catalytic activity. If this is the case, the white matter aspect of MNGIE will not be responsive to treatment strategies, which reduce or remove the pathological deoxyribonucleosides.

The adverse events for EE-TP observed in this compassionate study were nausea, coughing spasms, and erythema of the face and neck in two of the three patients. These events were not classified as serious and they were transient in nature, occurring within the first five minutes of EE-TP infusion. The events did not lead to patient discontinuation of EE-TP therapy, and they were managed by subsequent pre-medication with antihistamine, corticosteroid anti-inflammatory, and anti-emetic drugs. The introduction of a highly purified enzyme preparation allowed a careful and gradual withdrawal of this premedication. 

Recombinant biologicals are increasingly used as a treatment modality for diseases due to the deficiency of an enzyme or protein, and the development of antibodies against administered recombinant proteins is well documented [39]. Recombinant thymidine phosphorylase is encapsulated within autologous erythrocytes and it is less likely to elicit the strong antibody responses and neutralising antibodies that have been reported in the literature for other recombinant proteins. Antibodies that are specific to thymidine phosphorylase were detected in one patient, Patient 2 from treatment cycle 10, onwards. This observation does not raise any specific concerns with regard to neutralising antibodies, as the efficacy of encapsulated thymidine phosphorylase in metabolising the plasma metabolites improved over the 5.5 years of administration and positive clinical responses to treatment were recorded. The development of specific antibodies is not surprising when considering that senescent erythrocytes are naturally sequestered from the vascular compartment by macrophages of the monocyte-macrophage system, where macrophages are able to present peptides to T lymphocytes. Pre-clinical studies have demonstrated that the administration of erythrocyte encapsulated antigens to BALB/c mice is able to elicit humoral responses [40]. 

## 5. Conclusions

In conclusion, we have demonstrated the safety, tolerability, and efficacy of EE-TP in three patients with MNGIE under a compassionate use programme. Regulatory and ethical approval for a Phase 2 clinical trial of EE-TP has been granted to further investigate this treatment approach in patients with MNGIE.

## 6. Patents

Bax, B.E. and Bain, M.D. (2018) Treatment for mitrochondrial neurogastrointestinal encephalomyopathy (MNGIE). EP2760459B1.

## Figures and Tables

**Figure 1 jcm-08-00457-f001:**
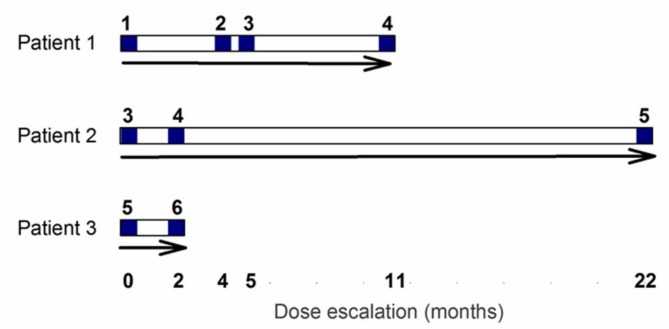
Dose escalation scheme for Patients 1, 2, and 3. Following a screening period of 28 days an intravenous dose of EE-TP was administered on day 0. Patients were escalated to the next dose following a safety review. Blue numbers refer to dose levels, where level 1 = 4 U/kg, level 2 = 9 U/kg, level 3 = 18 U/kg, level 4 = 29 U/kg, level 5 = 47 U/kg, and level 6 = 108 U/kg. Note that data pertaining to dose level 5 for Patient 2 is only covered in this report.

**Figure 2 jcm-08-00457-f002:**
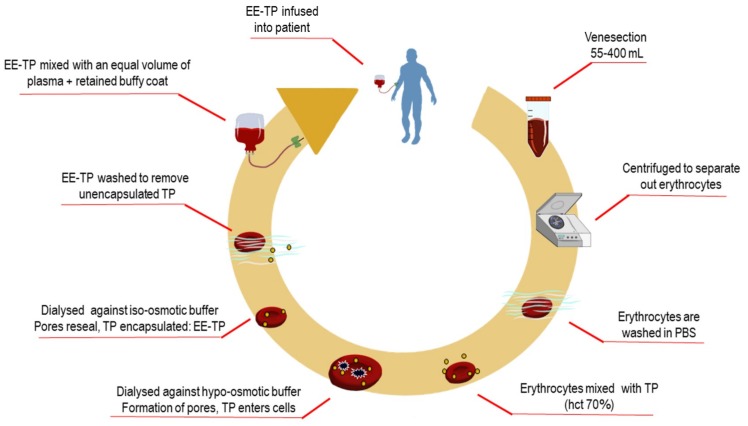
Manufacture of erythrocyte-encapsulated thymidine phosphorylase (EE-TP). Following venesection, blood was transferred to a Class A isolator for the manufacture of EE-TP under a Specials according to the Rules and Guidance for Pharmaceutical Manufacturers 2007 (MHRA). Blood was centrifuged to separate components and then the erythrocytes washed with phosphate buffered saline (PBS). Erythrocytes were then mixed with an appropriate activity of thymidine phosphorylase (TP) to a haematocrit of 70% and then dialysed against hypo-osmotic buffer for 90 min to create pores in the cell membrane. The lysed erythrocytes were then resealed by dialysis against iso-osmotic buffer for 60 min to encapsulate TP that had entered the cells. The resulting EE-TP was then washed to remove encapsulated TP, mixed with the retained buffy coat and an equal volume of autologous plasma, and then infused into the patient.

**Figure 3 jcm-08-00457-f003:**
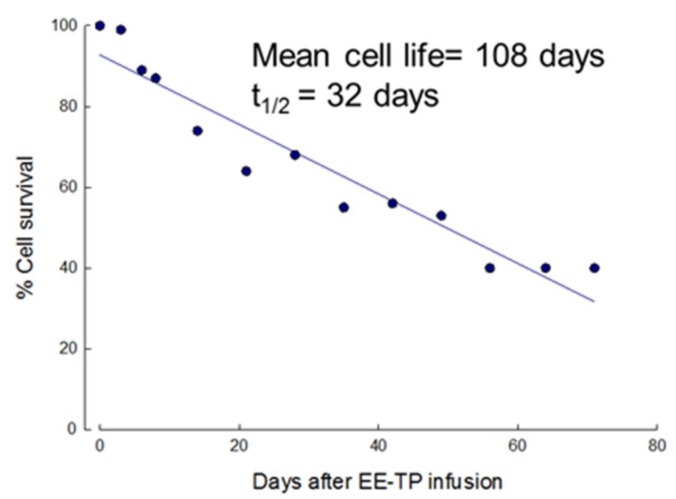
In vivo survival of EE-TP in Patient 1. The first administered dose of EE-TP (4 U/kg body weight) was labelled with ^51^Cr and the survival characteristics of the loaded erythrocytes assessed by following the disappearance of label from the circulation.

**Figure 4 jcm-08-00457-f004:**
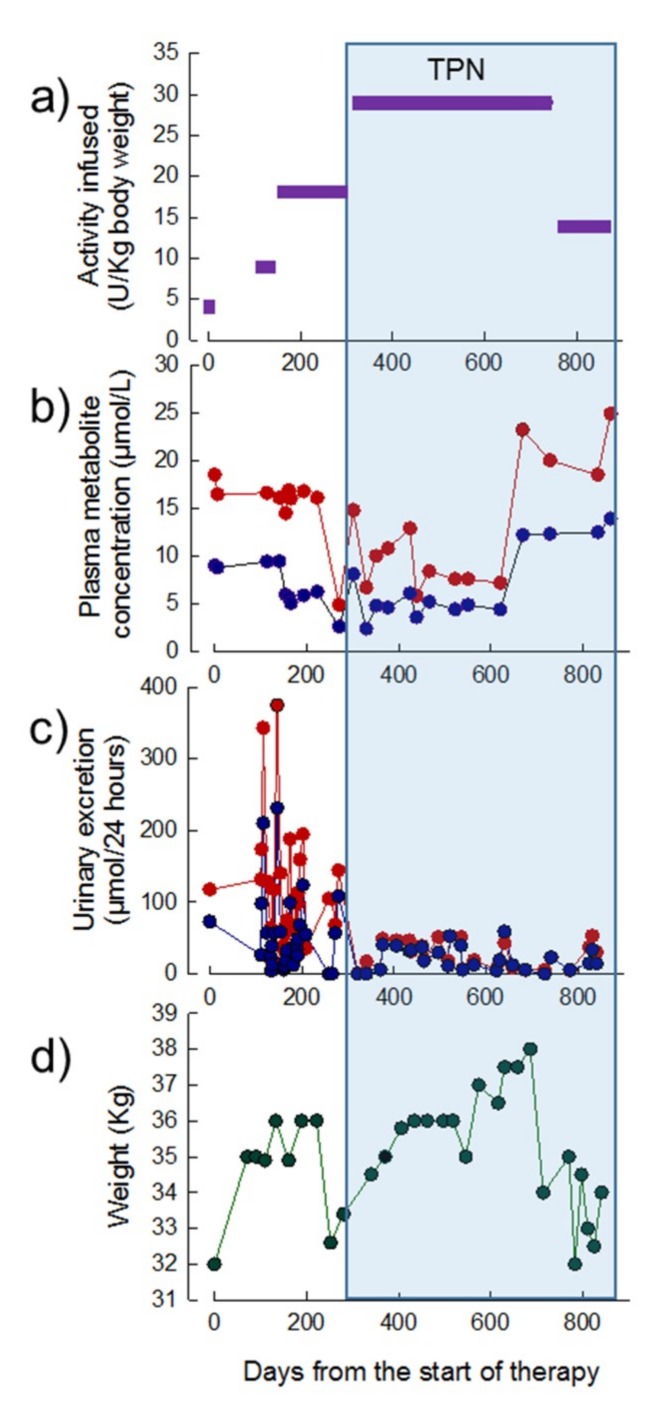
Dose of EE-TP administered, plasma concentrations of deoxyribonucleosides, urinary excretion of deoxyribonucleosides, and body weight in Patient 1 during 28 months of therapy with EE-TP. (**a**) The dose of encapsulated thymidine phosphorylase infused was determined using high performance liquid chromatography (HPLC). (**b**) Mid cycle plasma thymidine (blue data points) and deoxyuridine (red data points) concentrations were measured by Ultra-Performance Liquid Chromatography (UPLC). (**c**) Mid cycle urinary excretions of thymidine (blue data points) and deoxyuridine (red data points) were determined by measuring the concentration of metabolites in 24 h urine collections. (**d**) Patient body weight. The blue shaded area represents the timeframe in which TPN was administered.

**Figure 5 jcm-08-00457-f005:**
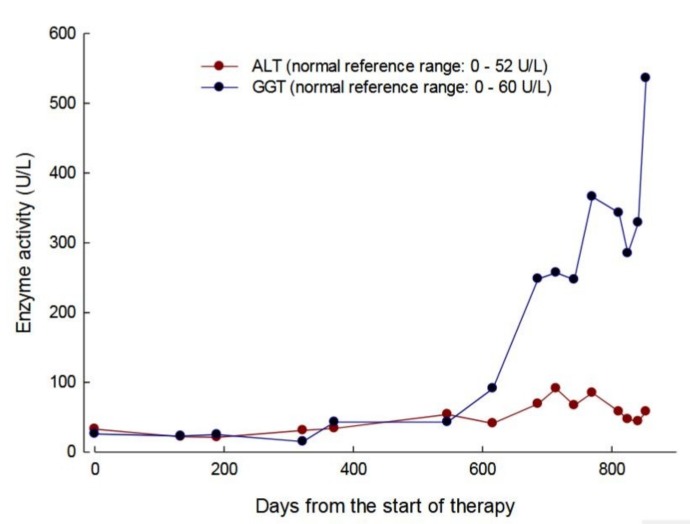
Serum alanine minotransferase (ALT) and gamma-glutamyl transpeptidase (GGT) activities in patient 1 during 31 treatment cycles of EE-TP administered over 28 months. The blue symbols represent GGT and red symbols ALT.

**Figure 6 jcm-08-00457-f006:**
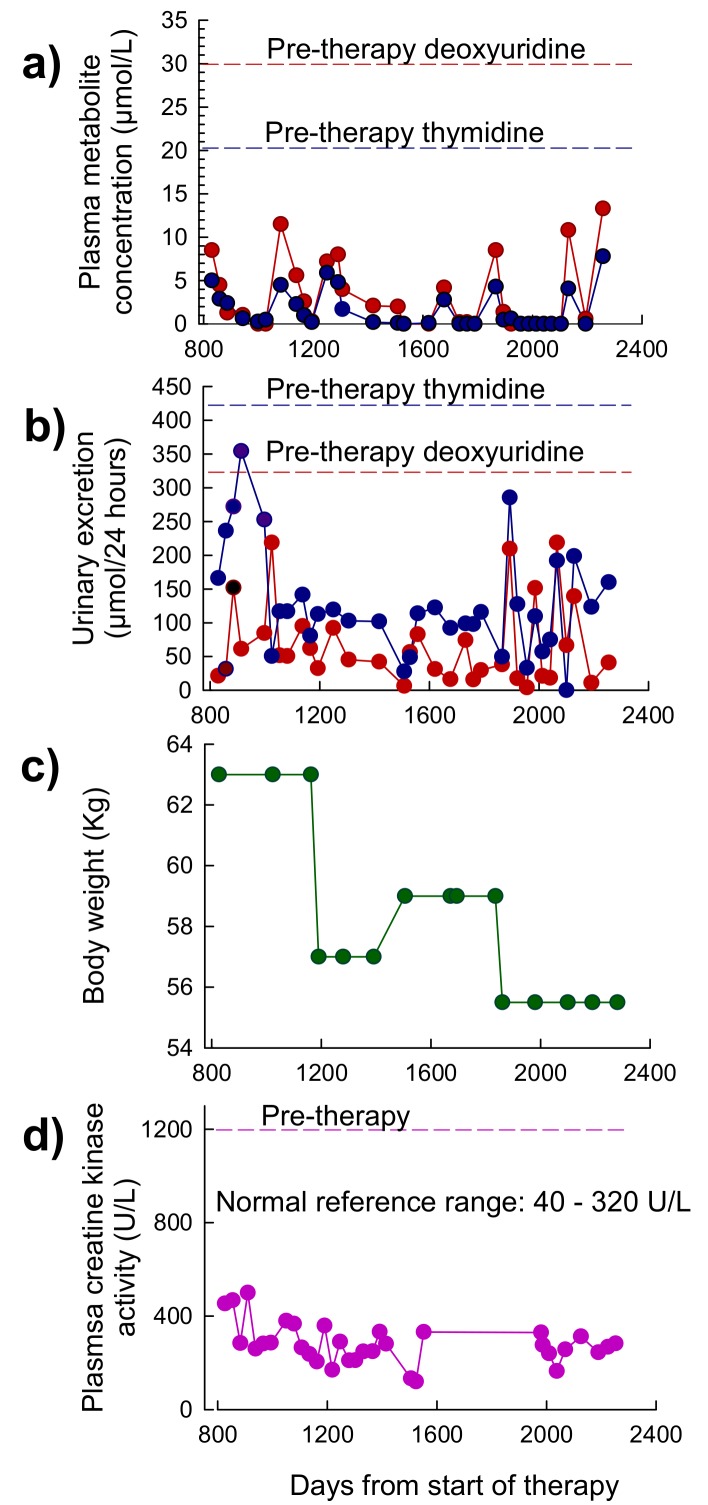
Plasma concentrations of deoxyribonucleosides, urinary excretion of deoxyribonucleosides, body weight and plasma creatine kinase in Patient 2 between 27 and 76 months of therapy with EE-TP. (**a**) Intra-cycle plasma thymidine and deoxyuridine concentrations were measured by UPLC. (**b**) Intra-cycle urinary excretions of thymidine and deoxyuridine were determined by measuring the concentration of the metabolites in 24 h urine collections. Red dotted lines represent pre-therapy deoxyuridine plasma concentration and urinary excretion, and blue dotted lines represent pre-therapy thymidine plasma concentrations and urinary excretion. (**c**) Patient body weight. (**d**) Plasma creatine kinase activity. Pink dotted line represents pre-therapy activity.

**Figure 7 jcm-08-00457-f007:**
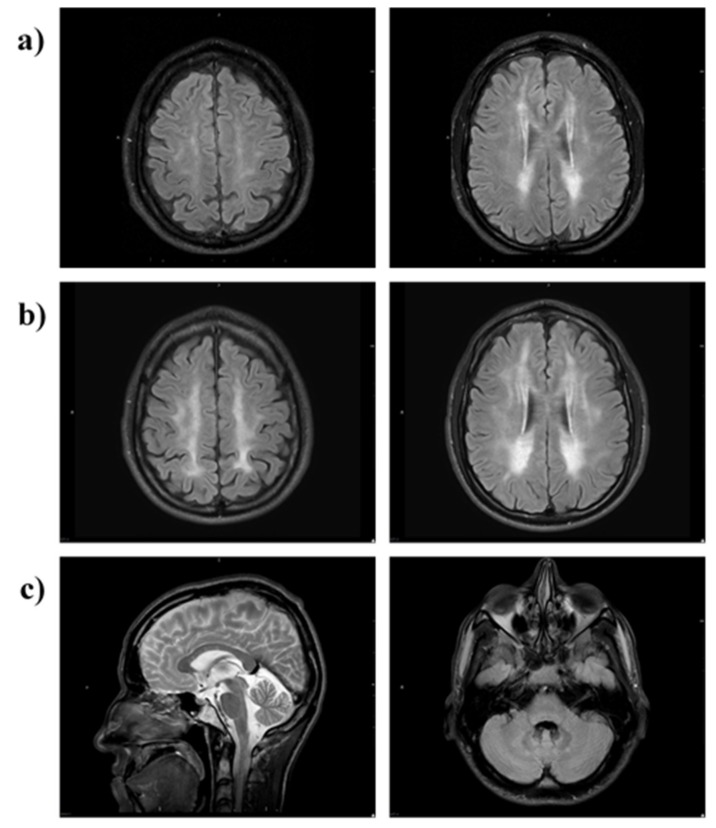
MRI brain of Patient 2. MRI (axial FLAIR) at (**a**) 28 months and (**b**) 63 months shows progression of T2/FLAIR hyperintense cerebral white matter changes (leukoencephalopathy). The scan at 28 months shows patchy involvement of cerebral periventricular, deep and subcortical white matter (**a**) which 35 months later is more confluent and extensive (**b**). There is relative sparing of the U-fibres and the corpus callosum (**c**, left image) and infratentorially there is minor symmetrical FLAIR hyperintensity in the cerebellar peridentate white matter (**c**, right image).

**Figure 8 jcm-08-00457-f008:**
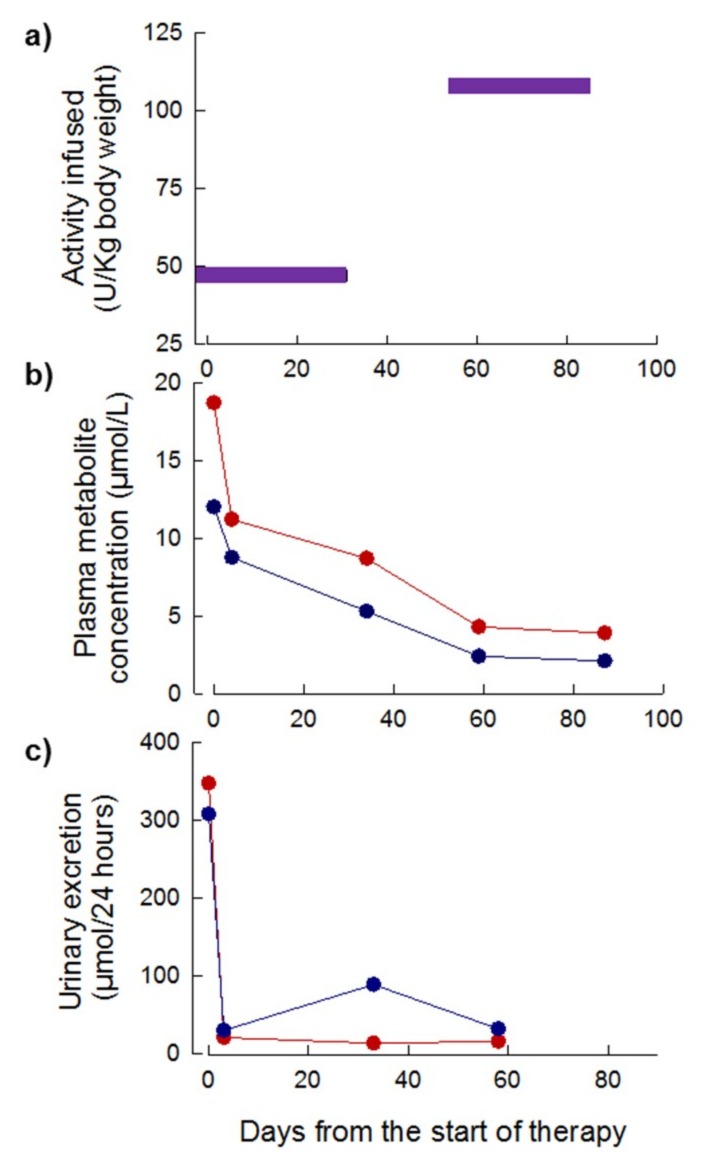
Dose of EE-TP administered, plasma concentrations of deoxyribonucleosides and urinary excretion of deoxyribonucleosides in Patient 3 during six months of therapy with EE-TP. (**a**) The dose of encapsulated thymidine phosphorylase infused was determined using HPLC. (**b**) Intra-cycle plasma thymidine (blue data points) and deoxyuridine (red data points) concentrations were measured by UPLC. (**c**) Intra-cycle urinary excretions of thymidine (blue data points) and deoxyuridine (red data points) were determined by measuring the concentration of metabolites in 24 h urine collections.

**Table 1 jcm-08-00457-t001:** Subject demographics and baseline disease characteristics.

Patient	Sex	Age of Diagnosis (Years)	Age at Start of EE-TP	*TYMP* Mutation	Buffy Coat TP Activity (nmol/mg Protein/h) ^1^	Pre-Treatment Plasma Metabolites (µmol/L)	
						Thymidine	Deoxyuridine
1	Female	23	25	Homozygous for c.1431_1432insT	40	9	19
2	Male	26	28	Heterozygous c.866A > C c.1231_1243 del	0	21	31
3	Male	15	26	g.4009_4010insG g.4101G > A	24	12	19

^1^ Normal range: 336–1341 nmol/h/mg Protein.

**Table 2 jcm-08-00457-t002:** Urinary excretion of ^51^Cr after infusion of EE-TP in Patient 1.

Urine Collection Period (h)	Urinary Excretion of ^51^Cr (%)	Reference Range (%)
0–24	1.1	1.0–3.2
24–48	0.8	
48–72	0.7	
Mean	0.9	

**Table 3 jcm-08-00457-t003:** Quality of Life and Disease scores for Patient 1, pre-therapy and during EE-TP therapy.

Scale	Months of Therapy		
	Pre-Therapy	7	24
SF36 (health and well-being):			
Physical component (population mean 50 ± 10)	47	52	42
Mental component (population mean 50 ± 10)	43	45	42
Newcastle mitochondrial disease scale (normal = 0):			
I (general neurological functioning, 0 to 50)	4	4	16
II (system specific functioning, 0 to 45)	6	6	6
III (clinical assessment, 0 to 50)	8	8	19
MRC sum motor score			
(normal = 80)	73	73	53
Overall neuropathy limitation scale:			
(0 = no disability, 12 = maximum disability)	4	4	4
Sensory sum score:			
(normal = 0, maximum score = 64)	26	26	34

**Table 4 jcm-08-00457-t004:** Quality of Life and Disease scores for Patient 2, pre-therapy and during therapy with EE-TP.

Scale	Months of Therapy				
	Pre-Therapy	23	50	64	73
SF36 (health and well-being)					
Physical component (population mean 50 ± 10)	48	52	55	55	55
Mental component (population mean 50 ± 10)	60	59	60	60	60
Newcastle mitochondrial disease scale (normal = 0)					
I (general neurological functioning, 0 to 50)	4	4	4	4	2
II (system specific functioning, 0 to 45)	2	2	2	6	5
III (clinical assessment, 0 to 50)	11	11	11	9	8
MRC sum motor score					
(normal = 80)	56	74 *	74	74	74
Overall neuropathy limitation scale:					
(0 = no disability, 12 = maximum disability)	3	3	3	4	4
Sensory sum score:					
(normal = 0, maximum score = 64)	21	19	20	13	4

* Previously reported [19].

**Table 5 jcm-08-00457-t005:** Adverse reactions to EE-TP administration in Patient 1.

Infusion #	Observations	Pre-Medication Prior to Infusion
2	nausea felt hot and faint erythema of face, neck and in arm proximal to infusion site tickle sensation in throat subsided spontaneous after 5 min	None
3	coughing spasm sensation in throat described as a tickle dry mouth felt hot subsided spontaneously after 5 min	Oral chlorphenamine 4 mg 3 h prior
4	coughing spasm	Oral chlorphenamine 8 mg 3 h prior
5	generalized paresthesia nausea coughing spasm generalized erythema transient headache, visual disturbance subsided spontaneously after 5 min	Oral chlorphenamine 8 mg 3 h prior IV hydrocortisone 50 mg at time of reaction
6	no reactions	Oral dexamethazone 4 mg 6 h prior Oral chlorphenamine 8 mg 3 h prior IV maxalon 10 mg immediately prior to infusion
9	coughing spasm nausea felt hot	As infusion 6
10	nausea	Oral dexamethazone 4 mg 6 h prior Oral chlorphenamine 8 mg 3 h prior Oral ondansetron 4 mg 3 h prior
15	coughing spasm nausea felt cold and shaky	As infusion 10
17	coughing spasm nausea erythema of neck	Oral dexamethazone 8 mg 6 h prior Oral chlorphenamine 12 mg in two divided doses: 6 h (8 mg) and 2 h (25%) prior Oral ondansetron 8 mg 3 h prior
18–31	no reactions	Oral dexamethazone 10 mg 6 h prior Oral chlorphenamine 12 mg two divided doses: 6 h (8 mg) and 2 h (4 mg) prior Oral ondansetron 16 mg total in two equal doses, 6 and 3 h prior

**Table 6 jcm-08-00457-t006:** Adverse reactions to EE-TP administration in Patient 2.

Infusion #	Observations	Pre-Medication Prior to Infusion
3	erythema of face felt hot	none
4	erythema of face felt hot	oral dexamethasone 4 mg 6 h prior oral chlorphenamine 12 mg, 4 mg and 8 mg given respectively 4 h and 1 h prior oral ondansetron 8 mg given 1 h prior
5	erythema of face felt hot	oral dexamethasone 10 mg 6 h prior oral chlorphenamine 12 mg, 4 mg and 8 mg given respectively 4 h and 1 h prior to infusion oral ondansetron 8 mg given 1 h prior
6	erythema of face felt hot tightness in chest	oral dexamethasone 12 mg 6 h prior oral chlorphenamine 12 mg, 4 mg and 8 mg given respectively 4 h and 2 h prior to infusion + IV chlorphenamine 8 mg at start of infusion oral ondansetron 8 mg given 1 h prior
7–8	felt hot erythema of face reduced compared to previous cycles	oral dexamethasone 10 mg 6 h prior IV hydrocortisone 100 mg at start of infusion oral chlorphenamine 12 mg, 2 h prior to infusion and IV 8 mg at start of infusion oral ondansetron 8 mg given 1 h prior
9	slight erythema of face felt hot	oral monteleukast 20 mg daily for 3 days prior oral dexamethasone 10 mg 6 h prior IV hydrocortisone 100 mg at start of infusion oral chlorphenamine 12 mg, 2 h prior to infusion and IV 8 mg at start of infusion oral ondansetron 6 mg 1 h prior
10	slight erythema of face felt hot	oral monteleukast 20 mg daily for 3 days prior to infusion oral ketotifen 2 mg daily in 2 equally divided doses for 4 days prior to infusion oral dexamethasone 10 mg 6 h prior oral chlorphenamine 12 mg, 2 h prior oral ondansetron 8 mg given 1 h prior IV hydrocortisone 100 mg at start of infusion. IV 8 mg at start of infusion
11	very slight erythema of face	as infusion 10
12–13	No reactions	oral dexamethasone 10 mg 6 h prior oral chlorphenamine 12 mg, 2 h prior oral ondansetron 4 mg given 1 h prior IV hydrocortisone 100 mg at start of infusion IV 8 mg at start of infusion
14–22	No reactions	phased withdrawal of pre-medication starting with ondansetron, then oral dexamethasone and chlorphenamine
23–26	No reactions	IV chlorphenamine 4 mg at start of infusion IV hydrocortisone 100 mg at start of infusion
27	very slight erythema of face	IV chlorphenamine 8 mg at start of infusion
28–39	No reactions until infusion 39 when during last 5 min of infusion: erythema of face, tightness in chest	IV chlorphenamine 8 mg at start of infusion IV hydrocortisone 50 mg at start of infusion
40–68	No reactions	oral dexamethasone 10 mg 5 h prior to infusion oral chlorphenamine 4 mg 4 h prior to infusion IV chlorphenamine 10 mg at start of infusion IV hydrocortisone 100 mg at start of infusion
69-	No reactions	oral dexamethasone 10 mg 5 h prior to infusion oral chlorphenamine 4 mg 4 h prior to infusion IV chlorphenamine 10 mg at start of infusion IV hydrocortisone 50 mg at start of infusion
70	No reactions	oral dexamethasone 10 mg 5 h prior to infusion oral chlorphenamine 4 mg 4 h prior to infusion IV chlorphenamine 10 mg at start of infusion
71	No reactions	IV chlorphenamine 5 mg at start of infusion IV hydrocortisone 50 mg at start of infusion
72	No reactions	oral dexamethasone 10 mg 4 h prior to infusion
73	No reactions	oral dexamethasone 6 mg 4 h prior to infusion
74–75	No reactions	oral dexamethasone 4 mg 4 h prior to infusion
76	Chill 3 h post infusion	none
77–79	No reactions	IV hydrocortisone 100 mg at start of infusion

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
