# Peer review of "Safety and Efficacy of Erythrocyte Encapsulated Thymidine Phosphorylase in Mitochondrial Neurogastrointestinal Encephalomyopathy"

_jcm, 2019, doi:10.3390/jcm8040457_

Reviewer 1 Report

This draft presented an effective pre-clinical trial that demonstrated the beneficial efficacy of EE-TP in 3 MNGIE patients over extended period, and observed minor and mostly manageable adverse reactions . The results are promising considering that for MNGIE and other mitchondrial diseases, there is quite  few treatments available yet. I am exited to read the positive result in patients. It is a please to read this draft because the experimental design, the protocol, the comprehensiveness are convincing, and the draft is beautifully well-written.

Author Response

Thank you for you review.

Reviewer 2 Report

This is a well-written report on a novel therapeutic option in a very rare disease designated mitochondrial neurogastrointestinal encephalomyopathy caused by thymidine phosphorylase deficiency.

The Authors are known in the field and had published their pilot study already - in the current report the safety and efficacy of encapsulated thymidine phosphorylase. While the methodology, data and their presentation are sound and original, some minor points should be addressed:

1) p. 11, 2nd para: Dynamics in MRI are not well understood in MNGIE, and MRI quantification is to the best of my Knowledge far from well-established/-validated - the current expert opinion is in the direction that probably no progression is typically seen in patients typically diagnosed with a full-blown clinical picture of MNGIE. However, while data on this subject are virtually absent, a discussion on putative TP function in the brain and blood-brain barrier in terms of the therapeutic approach presented is suggested. 

2) p.16, first para: In terms of immunogenicity: are there any correlations between pyrimidine serum levels and anti-thymidine phosphorylase antibodies. Please also discuss in more detail the presence of immunogenicity in terms of treatment response. 

3) p. 18, 2nd para: When discussing effects of total parenteral Nutrition (TPN) on treatment response, another putative aspect should also be addressed. Liver is (under physiological conditions) known to harbour significant TP production. In terms of TPN the liver becomes severely damaged (as reflected in cirrhosis considered a contraindication of allogenic stem cell transplantation in MNGIE), this may contibute to a breakdown of residual TP production in the liver in patients with MNGIE. Such reasoning could be added to the otherwise well-balanced discussion.

Author Response

The authors thank the reviewer for his/her constructive feedback and provide the following responses:

Comment 1.

Page 19, Line 652: Post mortem histopathological studies of brains have shown the presence of albumin in the cytoplasm of reactive astrocytes in patients with MNGIE, compared to age-matched healthy individuals, suggesting functional blood brain barrier alterations and consequent vasogenic oedema as a cause of leukoencephalopathy in MNGIE [34]. Furthermore, a mild perivascular gliosis was also observed in immunohistochemical analyses [35]. Thymidine phosphorylase is also referred to as gliostatin, and has been shown to exert  strong inhibitory effects on glial cells and neurotrophic effects on cortical neurons [36–38]. The leukoencephalopathy in MNGIE may therefore be a consequence of the absence of the inhibitory activity of thymidine phosphorylase, rather than its catalytic activity. If this is the case, the white matter aspect of MNGIE will not be responsive to treatment strategies which reduce or remove the pathological deoxyribonucleosides.

Comment 2.

Page 16 line 529: There were no correlations between the plasma levels of deoxyribonucleosides and the appearance of anti-thymidine phosphorylase antibodies.

Page 19, line 678: and positive clinical responses to treatment were recorded.

Comment 3.

Page 18, line 622: Thymidine phosphorylase is highly expressed in the liver of healthy individuals; for patients with MNGIE who express residual enzyme activity, the administration of TPN should be cautiously appraised.